# Emerging Highly Pathogenic Avian Influenza H5N1 Clade 2.3.4.4b Causes Neurological Disease and Mortality in Scavenging Ducks in Bangladesh

**DOI:** 10.3390/vetsci12080689

**Published:** 2025-07-23

**Authors:** Rokshana Parvin, Sumyea Binta Helal, Md Mohi Uddin, Shadia Tasnim, Md. Riabbel Hossain, Rupaida Akter Shila, Jahan Ara Begum, Mohammed Nooruzzaman, Ann Kathrin Ahrens, Timm Harder, Emdadul Haque Chowdhury

**Affiliations:** 1Department of Pathology, Faculty of Veterinary Science, Bangladesh Agricultural University, Mymensingh 2202, Bangladesh; sumyea1701030@bau.edu.bd (S.B.H.); mohiuddin1701119@bau.edu.bd (M.M.U.); sadia44282@bau.edu.bd (S.T.); riabbel.1701111@bau.edu.bd (M.R.H.); rupaida45771@bau.edu.bd (R.A.S.); jahan.begum@bau.edu.bd (J.A.B.); 2Department of Population Medicine and Diagnostic Sciences, College of Veterinary Medicine, Cornell University, Ithaca, NY 14853, USA; mn496@cornell.edu; 3Institute of Diagnostic Virology, Friedrich-Loeffler-Institute, Federal Research Institute for Animal Health, Suedufer 10, 17493 Greifswald, Germany; annkathrin.ahrens@fli.de (A.K.A.); timm.harder@fli.de (T.H.)

**Keywords:** HPAI H5N1, scavenging duck, Clade 2.3.4.4b, mutation, meningoencephalitis, encephalomalacia

## Abstract

Due to the global concern of H5N1 clade 2.3.4.4b in different host ranges, this study investigates the molecular epidemiology and pathology of HPAI H5N1 viruses in unvaccinated scavenging ducks in Bangladesh, with the goal of assessing viral evolution and associated disease outcomes. Ducks affected with HPAIV showed incoordination, torticollis, and paralysis. Pathological examination revealed prominent meningoencephalitis, encephalopathy, and encephalomalacia, along with widespread lesions in the trachea, lungs, liver, and spleen, suggestive of systemic HPAIV infection. A phylogenetic analysis of full-genome sequences confirmed continued circulation of clade 2.3.2.1a genotype G2 in these ducks. A mutation analysis of the HA protein in clade 2.3.4.4b viruses revealed potential antigenic drift and receptor-binding adaptation. The detection of clade 2.3.4.4b with marked neurological and systemic lesions suggests ongoing viral evolution with increased pathogenic potential also for ducks. These findings highlight the urgent need for enhanced surveillance and biosecurity to control HPAI spread in Bangladesh.

## 1. Introduction

The poultry industry in Bangladesh has faced significant threats from influenza A viruses (IAVs), posing serious challenges to both the national economy and public health. Since the first reported outbreak of highly pathogenic avian influenza virus (HPAIV) H5N1 in 2007, the virus has caused widespread damage [1,2]. HPAI H5N1 is known for its severe pathogenicity and broad host range, infecting various domestic birds such as chickens, ducks, and quail, but only rarely pigeons [3,4], as well as wild and migratory bird species [5]. HPAI viruses have been intermittently detected in Bangladesh, including clade 2.3.4.4b H5N6 in domestic poultry and clade 2.3.4.4h H5N6 in migratory birds and domestic ducks [6,7]. Phylogenetic analyses indicate that migratory birds traveling along the Central Asian flyway have played a key role in the transboundary spread of these virus strains [8]. Globally, the first outbreak of the AIV H5N1 subtype, belonging to the A/goose/Guangdong/1/96 (Gs/GD/96) lineage, caused an HPAI outbreak in poultry in China [9]. Since 2008, this novel reassortant H5 virus has been detected in ducks and live bird markets in China, carrying clade 2.3.4 hemagglutinin in combination with various neuraminidase subtypes such as N2, N5, N6, and N8. Over time, clade 2.3.4 further evolved into four genetically distinct subgroups [10]. According to the World Health Organization (WHO) classification, H5 viruses of clade 2.3.4.4 have evolved into eight distinct subclades, labeled 2.3.4.4a through 2.3.4.4h [11]. The H5N6 virus from subclade 2.3.4.4a was initially identified in poultry in Laos in 2013 and subsequently spread to Vietnam and China. The widespread distribution of H5N8 viruses from clade 2.3.4.4 across various regions and host species, along with recurrent outbreaks caused by clade 2.3.4.4b between 2016 and mid-2020 [12,13], led to the emergence of a novel H5N1 virus within clade 2.3.4.4b in late 2020. By the end of 2021, this H5N1 variant had largely replaced the circulating H5N8 strains of the same clade worldwide [14]. Between 2014 and 2017, the virus spread to North America, facilitating the development of a new clade, 2.3.4.4c [15]. H5N6 viruses from subclades 2.3.4.4d through 2.3.4.4h became established in both domestic and wild bird populations throughout Southeast Asia [16,17]. In Europe, an H5N8 virus from clade 2.3.4.4b detected in turkeys in Poland was genetically linked to Eurasian low-pathogenic avian influenza (LPAI) viruses and H5N8 viruses previously identified in sub-Saharan Africa [18]. Similar strains were subsequently reported in poultry and wild birds across several European countries [19,20]. Following their emergence, clade 2.3.4.4b H5N1 viruses have undergone extensive reassortment with LPAI viruses, leading to significant genotypic diversity and widespread dissemination across Europe, Africa, Asia, and the Americas [21,22,23].

In February 2007, H5N1 clade 2.2.2 was first detected in Bangladesh, followed by the emergence of clade 2.3.4.2 and clade 2.3.2.1a in 2011, affecting multiple host species [24] and circulating within live poultry markets [25]. By 2012, clades 2.2.2 and 2.3.4.2 were no longer detected, and clade 2.3.2.1a became the dominant strain associated with outbreaks in the country [25,26]. The reassortant virus has since maintained a stable presence, with ongoing detection in domestic poultry and live bird markets, indicating its continued circulation [27,28,29]. Unlike earlier H5N1 variants, recently emerged clade 2.3.4.4b viruses have demonstrated remarkable genetic plasticity, an expanded host range, and enhanced environmental persistence, contributing to sustained transmission across continents.

In Bangladesh, ducks are the second most significant contributors to national poultry meat and egg production [30]. As natural reservoirs and primary hosts of influenza A viruses, ducks play a pivotal role in the ecology, persistence, and transmission of HPAI H5N1 viruses [31]. The majority of domestic ducks belong to the species *Anas platyrhynchos* and are typically raised under scavenging or semi-intensive systems. These husbandry practices often bring ducks into close contact with other domestic poultry and wild migratory waterfowl, thereby increasing the risk of interspecies transmission of avian influenza viruses [32].

Given that ducks often remain asymptomatic while shedding high viral loads, they serve as silent reservoirs, facilitating the undetected spread of the virus to other domestic poultry species and across regions [33]. Scavenging ducks were selected as the focal host due to their unique ecological role in Bangladesh’s poultry production system, where they frequently interact with wild birds, poultry, and shared water bodies—making them important sentinels and amplifiers of AIV transmission. Additionally, the emergence of novel reassortant strains such as clade 2.3.4.4b necessitates continuous and enhanced genomic surveillance to monitor viral evolution, reassortment patterns, and potential zoonotic threats. Despite numerous outbreaks, comprehensive data on the current genetic makeup and transmission dynamics of HPAI viruses in ducks in Bangladesh remain limited. This gap hinders the development of effective control strategies, including biosecurity measures and targeted vaccination campaigns. Therefore, the current research focused on isolating and identifying H5N1 viruses from the scavenging ducks of Bangladesh. Full-genome sequencing and phylogenetic analysis of the isolated viruses, focusing on characterizing their genetic relationship to regional and global H5N1 strains, especially clade 2.3.4.4b, were performed. Gross and histopathological changes in infected ducks were documented to better understand the disease pathology and its contribution to transmission and mortality.

## 2. Materials and Methods

### 2.1. Ethics Statement

The scavenging ducks were handled as per the institution’s policies for the handling and use of laboratory animals. This study has been approved by the ethical committee of the Bangladesh Agricultural University Research System (protocol number: BAURES/ESRC/43/2024).

### 2.2. Sampling Collection and Processing

This study was conducted in different regions of the Mymensingh division of Bangladesh from June 2022 to March 2024. A total of 40 scavenging duck flocks were selected based on the presence of at least one reported respiratory or neurological clinical sign. These flocks, ranging in size from approximately 70 to 2500 ducks, were included as part of active and passive surveillance in high-risk areas. Necropsy was performed on site with appropriate safety measures and gross pathological changes were recorded. Pools of lung and tracheal tissues were collected aseptically from each duck at necropsy in sterile screw-capped tubes for virological analysis. Samples were stored at −80 °C until further analysis. For histopathology, tissues including lungs, trachea, brain, liver, cecal tonsil, and spleen were collected in 10% neutral buffered formalin. The formalin-fixed tissues were embedded in paraffin and sectioned for routine hematoxylin and eosin (H&E) staining. The H&E-stained sections were examined under a compound light microscope (Olympus BX43, Olympus Corporation, Tokyo, Japan).

### 2.3. Nucleic Acid Extraction

Pooled tissues of the lung and the trachea were collected in 2 mL of minimal essential medium supplemented with streptomycin and penicillin. A Tissue Lyser instrument (Qiagen, Hilden, Germany) was used to homogenize tissue samples for two minutes using a single 5 mm stainless steel bead in a 2 mL collection tube. Following centrifugation at 3000× *g* for 10 min, 200 µL of the clarified tissue homogenate was collected and used for nucleic acid extraction using the QIAampViral RNA Mini Kit (QIAGEN, Hilden, Germany).

### 2.4. Taq-Man Multi-Target RT-qPCR for Screening and Subtyping of Avian Influenza Virus

RT-qPCR was used to determine whether AIV was present in the samples. A previously established [34] TaqMan probe-based single-step RT-qPCR assay was employed for the detection of the M gene using AgPath Universal Probe One-Step RT-qPCR Kit (Thermo Fisher Scientific, Waltham, MA, USA). After that, all AIV M gene RT-qPCR-positive samples were subtyped to confirm the H5N1. Each RT-qPCR reaction was performed in a total volume of 12.5 µL, which included 2.5 µL of RNA template, 6 µL of 2X RT-PCR master mix, 0.5 µL of enzyme mix, 1.5 µL of nuclease-free water, and 2 µL of a primer-probe mixture (10 pmol of each component). The thermal cycling conditions include reverse transcription at 45 °C for 10 min, followed by an initial denaturation step at 95 °C for 10 min. This was followed by 40 cycles of denaturation at 95 °C for 15 s and annealing/extension at 60 °C for 1 min, during which fluorescence signals were captured. After the run, amplification plots were examined, and Ct (cycle threshold) values were determined. Ct values were used to estimate the relative viral load, and samples with Ct values of less than 35 were considered positive. The Optimized TaqMan Multiplex was used to simultaneously detect generic AIV (M gene) and a subtype of AIV (H5 and N1 genes). The false-negative and false-positive results were verified using the positive and negative control within each sample panel.

### 2.5. Genome Sequencing

Selected positive samples (based on the ct values ranging between 20 and 28) were considered for sequencing. As previously described, a nanopore-based sequencing method was utilized for whole-genome sequencing of HPAIV H5N1 [35]. Universal amplification of extracted RNA was performed using AIV-End-RT-PCR with the SuperScript III One-Step RT-PCR System (Thermo Fisher Scientific, Waltham, MA, USA). The resulting PCR products were purified using AMPure XP magnetic beads (Beckman-Coulter, Indianapolis, IN, USA). The whole-genome sequencing was carried out on the Mk1C MinION platform (Oxford Nanopore Technologies, Oxford, UK) using an R9.4.1 flow cell and the Rapid Barcoding Kit, following the manufacturer’s protocols. Consensus sequences were generated using a reference-based mapping approach with MiniMap2. Final genome assemblies were manually refined in Geneious Prime version 2025.0.3 (Biomatters, Auckland, New Zealand) to retain the highest-quality reads (≥60%).

### 2.6. Phylogenetic Analysis

Contemporary H5N1 sequences from Bangladesh, along with representative highly pathogenic avian influenza virus (HPAIV) field strains, were retrieved from public databases including the NCBI GenBank and GISAID. Multiple sequence alignments were performed using the online version of MAFFT (https://mafft.cbrc.jp/alignment/server/; accessed on 21 January 2025). Maximum likelihood trees were generated using the GTR model incorporated in MEGA XII software, with branch support assessed through 1000 bootstrap replicates. Final tree visualizations and annotations were completed using MEGA XII [36] and Inkscape 1.0 (https://inkscape.org; accessed on 25 January 2025). The obtained full-genome sequences of seven Bangladeshi H5N1, A/duck/Bangladesh/SD1/2022(SD1), A/duck/Bangladesh/SD4/2022(SD4), A/duck/Bangladesh/SD9/2023(SD9), A/duck/Bangladesh/SD13/2023(SD13) A/duck/Bangladesh/SD22/2023(SD22), A/duck/Bangladesh/SD34/2024(SD34), and A/duck/Bangladesh/SD35/2024(SD35), were deposited in the GISAID database with the accession number mentioned in Appendix A.

### 2.7. Molecular Analysis

Amino acid mutation profiles of potentially meaningful functional residues of HA, NA, and six internal proteins were compared with the selected Bangladeshi strains and reference clade sequences of H5N1 viruses using the Bioedit software version 7.2 (https://bioedit.software.informer.com/7.2/: accessed on 12 January 2025) and verified in the web-based tool Flu Surver incorporated into the GISAID platform. The cleavage site, receptor-binding sites (RBSs), and glycosylation sites were given special attention because they are critical determinants of the pathogenicity, host specificity, transmissibility, and immune evasion of AIVs.

## 3. Results

### 3.1. Clinical Features and Gross Lesions

From June 2022 to March 2024, 40 scavenging duck farms were monitored in the Mymensingh division, in northern Bangladesh. Ducks exhibiting clinical signs, primarily respiratory and neurological, were included in the study. Flock histories, including age, mortality rates, and duck plague vaccination status, were recorded (Table 1). Most of the affected ducks of 50% flocks (20/40) showed similar clinical signs and symptoms, which were mostly sudden death, torticollis (Figure 1A), drowsiness, paralysis, neck bending, coughing, convulsion, and rattling. Severely affected ducks were necropsied either on-site or at the host laboratory. The common gross pathological lesions found included congestion of the entire body of the duck and hemorrhages, which are considered signs of a systemic infectious disease in ducks. The significant postmortem lesions observed were severe fibrino-hemorrhagic pneumonia (Figure 1B) and hemorrhagic brain (Figure 1C). No significant gross lesion was found in other visceral organs. The severely affected organs were collected for molecular detection and fixed in 10% neutral buffered formalin for histopathological analysis.

### 3.2. Detection of Highly Pathogenic Avian Influenza Virus

HPAI H5N1 was detected in 14 of 40 flocks by RT-qPCR, with Ct values ranging from 19.73 to 33.47. The remaining 26 flocks tested negative for avian influenza virus (AIV), though the presence of other pathogens was not assessed here. A subset of H5N1-positive samples was selected for whole-genome sequencing based on low Ct values (19.73–26.39) and molecular characterization. Edited and assembled sequences were used for the construction of the phylogenetic tree.

### 3.3. Histopathological Lesions

In the affected ducks, different histopathological lesions were observed. In the trachea, the mucosal layer was found thickened due to inflammation (Figure 2A), with profuse mucous secretion and accumulation over or along the mucosal layer (Figure 2B). The inflammatory cells were predominantly lymphocytes. Histopathological examination of the lung revealed pleuritis with the loss of mucosal epithelium (Figure 2C) as one of the predominant lesions, along with pneumonia characterized by marked congestion and hemorrhages, alveolitis with thickened interalveolar septa, diffuse alveolar collapse, a focal loss of pneumocyte I (Figure 2D), and compensatory proliferation of pneumocyte II (Figure 2D: inset box).

HPAI-infected ducks showed significant pathological lesions in the brain. Most of the HPAI-positive ducks exhibited notable congestion, microgliosis, and numerous vacuoles within the brain parenchyma (Figure 3A). Additionally, severe focal to diffuse meningitis characterized by predominant mononuclear inflammatory cell infiltration was observed in many of the infected ducks (Figure 3B). In some cases, encephalitis (Figure 3C, circled) was evident, along with multifocal degenerative (encephalopathy) and necrotic foci indicative of encephalomalacia (Figure 3D, circled). Discrete necrosis of neurons accompanied by neuronophagia (Figure 3D, inset box) was also noted.

Other signs of systemic infection were observed in the liver: chronic multifocal necrotizing hepatitis in portal area (Figure 4A) and in liver parenchyma (Figure 4B). In the cecal tonsil, the main pathological lesions were typhlitis (Figure 4C). Additionally, we found muscular degeneration in the inner muscle layer of the cecal tonsil (Figure 4C, circle). In the spleen, the main pathological lesions observed were mild lymphoid depletion (Figure 4D).

In summary, HPAI H5N1-infected ducks showed predominant mononuclear inflammatory infiltration across multiple tissues. The significant lesions were observed in the respiratory central nervous and gastrointestinal systems. The respiratory tract showed tracheitis, pleuritis, and pneumonia. In the central nervous system, meningoencephalitis was evident along with encephalopathy, encephalomalacia, and neuronal necrosis accompanied by neuronophagia. As part of our field investigation, we did perform necropsy on apparently healthy ducks from non-infected flocks. These ducks showed no clinical signs or gross pathological lesions in the respiratory tract or other systemic organs during postmortem examinations. Since no gross abnormalities were observed, we did not proceed with a histopathological analysis of these tissues.

### 3.4. Phylogenetic Relationships

A total of seven out of fourteen positive samples were sequenced successfully. The phylogenetic trees were constructed for all eight genome sequences. The HA phylogenetic analysis revealed that five strains (SD1, SD9, SD22, SD34, and SD35) of H5N1 belonged to clade 2.3.2.1a, and the other two strains (SD4 and SD13) belonged to clade 2.3.4.4b of the Gs/Gd lineage (Figure 5). The strains of clade 2.3.2.1a are closely related to previous and contemporary H5N1 viruses of Bangladesh that have been circulating for nearly a decade. Importantly, two new strains of clade 2.3.4.4b were detected that are closely related to Korean, Vietnamese, and Japanese strains of H5N1. Previously isolated Bangladeshi clade 2.3.4.4b strains were also clustered with the current strains, indicating the continuing presence of clade 2.3.4.4b in Bangladesh. The NA sequences reflect the clade-specific clustering of the HA gene (Figure 5).

The six internal genes of the studied viruses from Bangladesh clustered respective to specified clades (Appendix A). However, the two 2.3.4.4b viruses detected in this study are distinct from a new Japanese cluster 2.3.4.4b isolated from a crow in Japan in 2024. The reassortment of internal genes among different clades or subtypes was not observed.

### 3.5. Mutational Analysis

Among seven H5N1 viruses, two distinct multibasic hemagglutinin (HA) cleavage site motifs were identified: PQKERRRKR*GLF, characteristic of clade 2.3.2.1a, and PLREKRRKR*GLF, associated with clade 2.3.4.4b. Both motifs are indicative of high pathogenicity, confirming the virulent nature of these clades. The receptor-binding domain (RBD) of the HA protein comprises the 130 loop, the 190 helix, and the 220 loop [37]. Within the receptor-binding site (RBS), the conserved residues Q226 and G228 (H3 numbering) remained unchanged across all isolates, suggesting a continued preference for avian-type receptors (sialic acid with α2,3-Gal linkage). However, other amino acid variations were observed within the 220 loop compared to the Gs/Gd strains of H5N1. In five of the seven viruses from clade 2.3.2.1a, mutations Q222K and N223I were detected. In contrast, two isolates belonging to clade 2.3.4.4b exhibited N223V and S227R substitutions. These mutations may reflect clade-specific adaptations and could influence receptor-binding efficiency or host ranges [38]. The receptor-binding pocket formed by the 190 helix, which has been associated with the HPAI phenotype in chickens [39], contains key mutations, such as E193R/K/N and Q196K. Among the seven H5N1 isolates studied, five viruses from clade 2.3.2.1a exhibited the E193R substitution, while two other isolates of clade 2.3.4.4b showed an E193N mutation. Additionally, the Q196K substitution was identified in two clade 2.3.4.4b viruses, further indicating potential functional divergence within the receptor-binding pocket among different clades. An S137A mutation within the 130 loop was consistently observed across all seven characterized H5N1 viruses. Overall, the study observed mutations such as T156A, S141P, and E193R/K in some of the strains. These substitutions are located within or near known antigenic sites and the receptor-binding domain and have been reported in previous studies to be associated with antigenic variation or altered receptor affinity [39,40,41]. The hemagglutinin (HA) protein contains several N-linked glycosylation sites, defined by the N-X-S/T motif (where X is any amino acid except proline), as well as multiple antigenic sites [40,41]. The number, type, and position of these glycan modifications vary among virus strains, potentially influencing host immune recognition and viral fitness [42]. However, the potential phenotypic impacts of these mutations require further investigation through in vitro receptor-binding assays, cell culture studies, or animal models to confirm their functional significance.

All studied H5N1 isolates are predicted to carry four N-linked glycosylation sites (H5 numbering) within the HA1 protein at positions 27 (NNS), 39 (NVT), 181 (NNT), and 302 (NNS), as well as one site in the HA2 protein at position 500 (NGT). Notably, the glycosylation site at positions 154–156 (NNT, H3 numbering)—commonly observed in other gs/GD strains—was absent in all isolates, instead showing the motifs DDA or DNA (Table 2). Additionally, an S141P mutation was identified in the antigenic site A (residues 136–141) of two recent clade 2.3.4.4b isolates. These variations could influence antigenicity and immune evasion [43]. A summary of the key HA amino acid residues is provided in Table 2.

In addition to the HA protein, an analysis of the internal genes PB2, PB1, PA, NP, NA, M, and NS revealed several conserved mutations known to influence viral replication and host adaptation. The mutations observed in these gene segments were consistent with those previously reported in our earlier characterization of clade 2.3.2.1a H5N1 isolates [28], suggesting a conserved genetic backbone within this clade. Notably, the signature PB2-E627K mutation, which enhances polymerase activity and viral replication in mammalian hosts [44], was not detected in our study isolates. Similarly, the PB1-F2 N66S substitution, known to increase virulence in mice [45], and the PA-T97I mutation, associated with increased replication efficiency [46], were also absent in the viruses from both clades 2.3.2.1a and 2.3.4.4b. These findings suggest that while these mutations are typically involved in viral adaptation to mammals, the viruses in this study appear to be primarily adapted to avian hosts, supporting their limited zoonotic potential. These findings also support the idea that while HA undergoes more rapid evolution for antigenic escape, the internal proteins maintain stability to preserve replicative fitness across various avian host species.

## 4. Discussion

Domestic free-range ducks can act as reservoirs for HPAIV because they are often not associated with the development of overt clinical disease. However, recent increases in duck mortalities in Bangladesh [47] accompanied by neurological signs due to HPAIV H5 infections suggests that ducks may no longer serve solely as reservoir hosts. In Bangladesh, domestic ducks form a bridge between migratory birds and poultry sold at live bird markets (LBMs), thereby creating an ideal environment for viral genetic evolution and interspecies transmission. In this study, we detected and characterized clade 2.3.4.4b and 2.3.2.1a HPAI H5N1 virus in samples from AIV surveillance in domestic duck flocks in Bangladesh, in 2022–2024.

The H5N1 clade 2.3.4.4b was detected in domestic scavenging duck flocks that were clinically affected and brought for postmortem analysis. The virus is genetically linked to the previous Bangladeshi H5N1 clade 2.3.4.4b detected in domestic free-range ducks that were clinically healthy [48]. However, the observed increase in pathogenicity, as well as the clinical infections and mortality seen in these scavenging ducks, despite sharing a similar genotype, remains unexplained. Further investigation is required to determine the factors contributing to this shift in virulence and its impact on both animal and human health.

The H5N1 clade 2.3.4.4b has raised significant concern worldwide due to its observed virulence, high transmissibility, and its ability to infect a range of hosts, including both avian and mammalian species. In the USA, the H5N1 clade 2.3.4.4b virus has caused a series of alarming spillover events, highlighting the growing risk of zoonotic transmission and mammalian adaptation [49,50]. In contrast, Bangladesh has experienced an endemic circulation of H5N1 viruses belonging to clade 2.3.2.1a, while clade 2.3.4.4b has predominantly spread across Asia and Europe. The clade 2.3.4.4b was first detected in domestic free-range ducks in Bangladesh, where it was initially reported to be clinically asymptomatic [48]. The detection of the HPAI H5N1 clade 2.3.4.4b virus in domestic scavenging ducks exhibiting clinical signs of infection underscores a shift in viral behavior that warrants closer attention. Their free-ranging behavior and lack of vaccination increase their susceptibility to infection and potential to serve as reservoirs or mixing vessels for novel AIV strains. Only two flocks, SD4 and SD13, were found positive for 2.3.4.4b, and their mortalities were 17.33% and 0.33%, respectively. The other clade 2.3.2.1 also has a border range of mortality as mentioned in Table 1. While a few H5N1-negative flocks showed relatively high mortality, we suspect this may be due to other concurrent infections common in scavenging duck populations, such as duck plague virus, Mycoplasma spp., Pasteurella multocida, or Riemerella anatipestifer. However, as the primary objective of this study was to investigate the presence and impact of HPAI H5N1, further diagnostic workup for other pathogens was not conducted.

Multiple experimental infection studies have demonstrated that HPAI infection in ducks can lead to severe clinical manifestations [51,52,53]. Research indicates that younger and meat-type ducks tend to exhibit higher virus loads in tissues, increased viral shedding, and elevated mortality rates ranging from 30% to 50%, often accompanied by pronounced neurological signs following HPAI infection [54,55,56]. The heightened susceptibility in younger ducks may be attributed to age-dependent variations in innate immune responses triggered by HPAI virus infection [56]. In our study, ducks from multiple age groups exhibited severe neurological signs, including head tremors and paresis, as detailed in Table 1. Correspondingly, histological examination revealed severe encephalitis, encephalomalacia, and neuronal necrosis, along with diffuse vacuolation in brain’s spongy matter (encephalopathy). In addition to central nervous system lesions, pronounced pathological lesions were observed in the respiratory and gastrointestinal systems, indicating a high level of pathogenicity in naturally infected ducks. Some of the lesions observed in the respiratory and gastrointestinal systems may have been influenced by poor management practices and co-infection with other viral or secondary bacterial infections, as previously reported in studies from Korea and Egypt [51,55]. This observation emphasizes a potentially increased pathogenicity of clade 2.3.4.4b in ducks. However, further controlled studies and broader surveillance including asymptomatic populations are needed to validate this observation.

In terms of genetic features, the surface glycoproteins, particularly HA, exhibit rapid evolution, primarily driven by immune selection pressure. These changes may support effective immune evasion and enhance viral replication efficiency in avian hosts, with the potential for adaptation to mammalian hosts as well. In contrast, the internal gene segments of the H5N1 viruses, particularly those belonging to clade 2.3.2.1a and 2.3.4.4b, circulating in Bangladesh, showed minimal evolution. This evolutionary conservation highlights the importance of ongoing genomic surveillance and functional studies to assess the pandemic risk posed by these circulating strains. Given all the local farm-level factors, including biosecurity gaps, free-range duck rearing, mixed-species farming systems, live poultry trade, and frequent and intense exposure to migratory wild birds, which would facilitate further genetic evolution and increase the risk of host spillover events, the seemingly stable genetic properties of HPAIV circulating in Bangladesh is astonishing and cannot be readily explained. In other regions where clade 2.3.4.4b virus circulates enzootically, i.e., in Europe and the Americas, a plethora of more than 200 genotypes have developed, which come and go [57].

## 5. Conclusions

The HPAI H5N1 clade 2.3.2.1a and 2.3.4.4b viruses continue to circulate in duck populations in Bangladesh. Mutation analysis of the HA protein in clade 2.3.4.4b viruses identified key amino acid substitutions affecting glycosylation site, antigenic site A, and the receptor-binding pocket, suggesting ongoing antigenic drift and adaptation in the receptor-binding domain. Infected ducks exhibited significant neuronal lesions, and the observed mortality indicates a possible increase in pathogenicity in this host species. These findings point to the continuing evolution of HPAI viruses with potentially enhanced virulence in ducks. The detection of molecular changes associated with antigenic drift and increased pathogenicity in circulating H5N1 viruses underscores the need for sustained genomic surveillance in avian hosts. Functional studies are essential to assess the implications of these mutations for host adaptation, transmission dynamics, and zoonotic potential. Strengthening active monitoring and preparedness is critical to mitigate the risks posed by evolving HPAI viruses in endemic regions.

## Figures and Tables

**Figure 1 vetsci-12-00689-f001:**
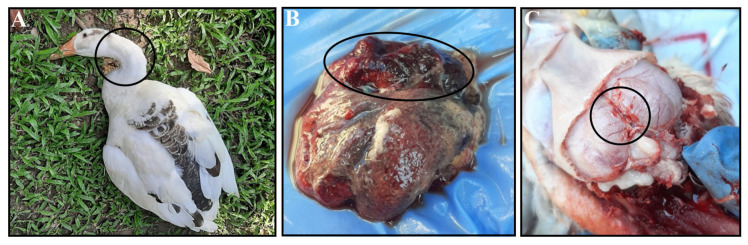
Clinical and postmortem findings in the affected ducks. The affected duck shows (**A**) torticollis, (**B**) severe fibrino-hemorrhagic pneumonia, and (**C**) hemorrhagic brain. All abnormalities are marked with black circles.

**Figure 2 vetsci-12-00689-f002:**
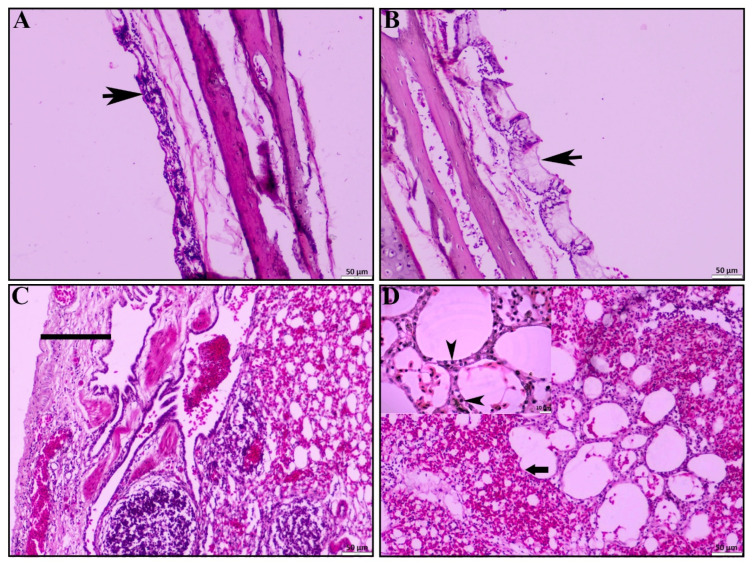
Histopathological lesions of trachea and lungs. (**A**) Tracheitis characterized by infiltration of reactive cells, predominantly mononuclear cells (black arrowhead). (**B**) Tracheitis with profuse mucous secretion (black arrowhead). (**C**) Pleuritis with pleural thickening (black line). (**D**) Severe pulmonary hemorrhages with loss of type I pneumocytes and compensatory proliferation of type II pneumocytes (inset, black arrowhead); H & E stain; scale bar indicates magnification.

**Figure 3 vetsci-12-00689-f003:**
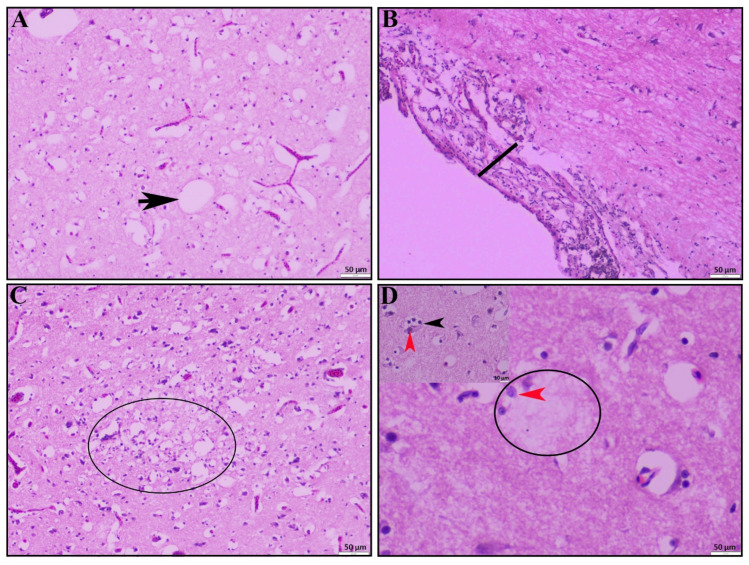
Histopathological lesions of brain. (**A**) Severe congestion and numerous vacuoles (black arrow) within brain parenchyma. (**B**) Meningoencephalitis characterized by marked thickening of the meninges. (**C**) Focal to diffuse encephalitis with encephalomalacia (circle) and (**D**) encephalomalacia (circle) accompanied by neuronophagia (inset, black arrowhead), and neuronal necrosis (red arrowhead); H & E stain; scale bar indicates magnification.

**Figure 4 vetsci-12-00689-f004:**
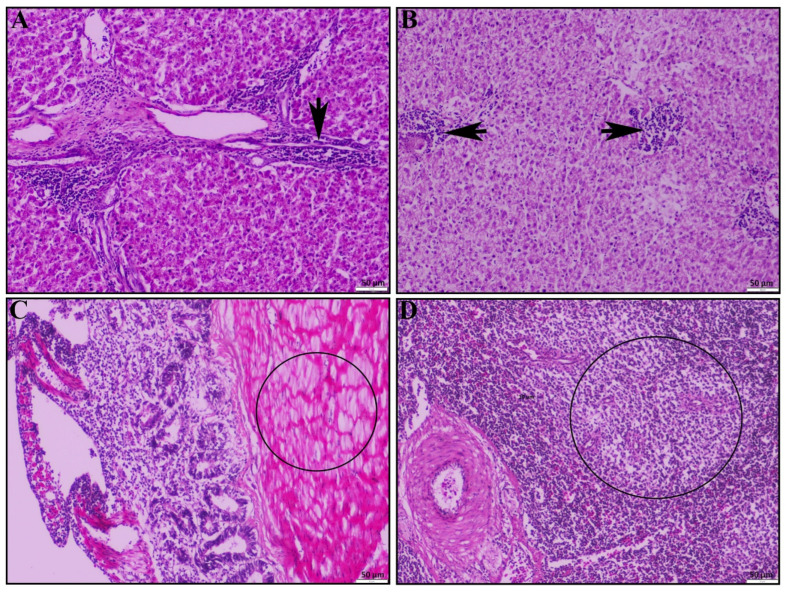
Histopathological lesions of the liver, cecal tonsil, and spleen of the dead duck. (**A**) Liver shows chronic necrotizing multifocal portal to peri-portal hepatitis (black arrow) and (**B**) inflammatory infiltrations in the liver parenchyma (black arrow). (**C**) Cecal tonsil shows typhlitis and muscular degeneration in inner muscular layer (circle). (**D**) The spleen shows mild lymphoid depletion (circle); H & E stain; scale bar indicates magnification.

**Figure 5 vetsci-12-00689-f005:**
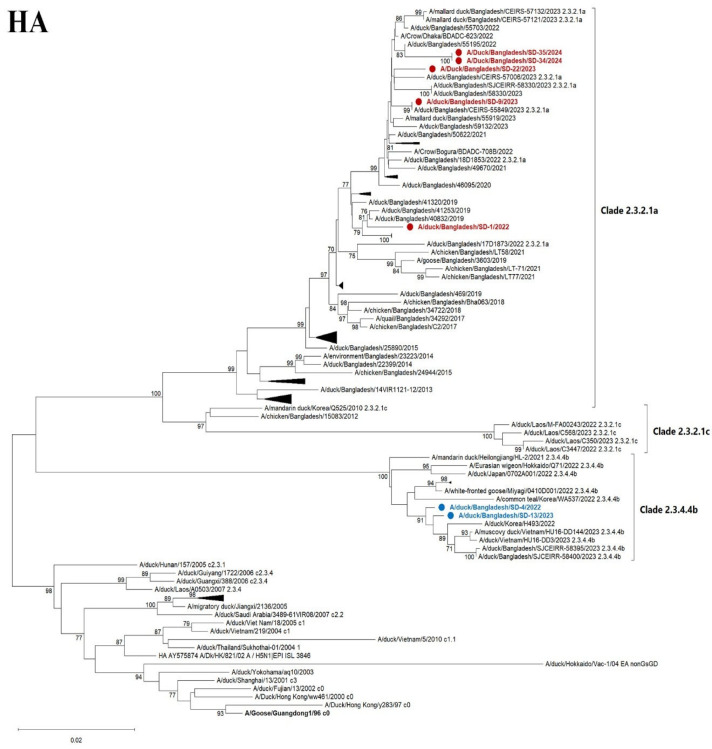
Maximum likelihood phylogenetic tree based on complete HA and NA genes. HA gene sequences of 102 H5N1 viruses (upper) and NA genes of gene sequences of 101 H5N1 (lower) of contemporary and representative strains. The clade names shown (e.g., 2.3.4.4b and 2.3.2.1a) refer to the hemagglutinin (HA) clade designation and are used here to indicate the overall genetic background of the viruses, which share both HA and NA segment relationships. The evolutionary history was inferred utilizing the maximum likelihood method based on the general time-reversible model with 1000 bootstrap replicates and visualized using MEGA XII software. The red color indicates 5 recently characterized strains of clade 2.3.2.1a, while the blue color indicates the 2 strains of clade 2.3.4.4b found in Bangladesh. Some other strains of the clades within the branch were collapsed.

**Table 1 vetsci-12-00689-t001:** Summarized flock history of the studied duck populations.

Shed/Flock ID	Age (Days)	Number of Birds	Number of Dead Birds	Mortality (%)	Vaccination	H5N1 RT-qPCR (Ct Value)
K4	120	200	0	0	No	
K7	38	2200	200	9.09	No	
K8	30	500	2	0.4	No	30.27
**SD1**	**90**	**300**	**10**	**3.33**	**Duck Plague**	**23.21**
SD2	60	1500	400	26.66	Duck Plague	
SD3	38	2200	200	9.09	No	
**SD4**	**13**	**75**	**13**	**17.33**	**No**	**19.73**
SD5	27	250	6	2.4	No	30.26
SD7	300	1150	40	3.48	Duck Plague	
SD8	210	2000	0	0	No	
**SD9**	**180**	**1200**	**0**	**0**	**No**	**23.45**
SD10	365	500	0	0	No	
SD11	150	800	0	0	No	
SD12	240	2500	1	0.04	Duck Plague	29.89
**SD13**	**210**	**1500**	**5**	**0.33**	**Duck Plague**	**25.96**
SD14	300	2000	3	0.15	Duck Plague	
SD15	540	1000	0	0	No	
SD16	75	300	0	0	No	30.95
SD17	150	120	0	0	No	
SD18	90	150	20	13.33	Duck Plague	32.06
SD19	420	120	11	9.17	No	
SD20	365	250	7	2.8	No	
SD21	210	2500	50	2	Duck Plague	
**SD22**	**45**	**600**	**45**	**7.5**	**Duck Plague**	**26.39**
SD23	250	400	25	6.25	No	
SD24	45	1000	300	30	Duck Plague	33.47
SD25	365	320	5	1.56	No	
SD26	40	550	200	36.36	Duck Plague	29.86
SD27	90	1000	200	20	Duck Plague	
SD28	60	300	50	16.66	No	
SD29	150	7	0	0	No	
SD31	90	550	10	1.82	Duck Plague, Cholera	
SD32	30	800	125	15.6	Duck Plague	
SD33 (a–e)	36	1600	800	50	Duck Plague	
**SD34**	**270**	**250**	**2**	**0.8**	**Duck Plague**	**24.48**
**SD35**	**45**	**52**	**28**	**53.85**	**Duck Plague**	**22.50**

Bold face samples are the positive H5N1 strains that have sequenced; (a–e) indicates the average of the five different flocks of a single farm.

**Table 2 vetsci-12-00689-t002:** Potential residual sites of HA protein of the studied H5N1 Bangladeshi strains.

H5N1 Isolates in This Study↓	Clade	HA Cleavage Site	190 Helix	130 Loop	220 Loop and RBS	N-Linked Glycosylation Site	Antigenic Site A
Positions-→ (H3 Numbering)		323–330	190–198	135–138	221–228	154–156	141
Gs-Gd/1996	1	PQRERRRKKR*GLF	EQTKLYQNP	VSSA	PKVNGQSG	NSA	S
SD4/2022	2.3.4.4b	PLREKRRK-R*GLF	EQTNLYKNP	VSAA	SQVNGQRG	DDA	P
SD13/2023		PLREKRRK-R*GLF	EQTNLYKNP	VSAA	SQVNGQRG	DDA	P
SD1/2022	2.3.2.1a	PQKERRRK-R*GLF	EQTRLYQNP	VSAA	SKINGQSG	DNA	S
SD9/2023		PQKERRRK-R*GLF	EQTRLYQNP	VSAA	SKINGQSG	DNA	S
SD22/2023		PQKERRRK-R*GLF	EQTRLYQNP	VSAA	SKINGQSG	DNA	S
SD34/2024		PQKERRRK-R*GLF	EQTRLYQNP	VSAA	SKINGQSG	DNA	S
SD35/2024		PQKERRRK-R*GLF	EQTRLYQNP	VSAA	SKINGQSG	DNA	S

Down arrow indicated H5N1 isolates name, Right arrow indicated position of the aminocid.

## Data Availability

The data used in the analyses were deposited in the GISAID platform (Accession numbers).

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
