# Peer review of "Emerging Highly Pathogenic Avian Influenza H5N1 Clade 2.3.4.4b Causes Neurological Disease and Mortality in Scavenging Ducks in Bangladesh"

_vetsci, 2025, doi:10.3390/vetsci12080689_

Round 1

Reviewer 1 Report

Comments and Suggestions for Authors

The manuscript [vetsci-3700947], entitled “Emerging highly pathogenic avian influenza H5N1 clade 2.3.4.4b causes neurological disease and mortality in scavenging ducks in Bangladesh” by Prof. Chowdhury, et al., reports the molecular epidemiology and pathology of 22

HPAI H5N1 viruses in unvaccinated scavenging ducks in Bangladesh.

This is an interesting and important study.

Some concerns are listed below for consideration in revision.

Some specific concerns:

  1. Abstract, the background section in the abstract is excessively lengthy, and the results section fails to specifically present the key data of this study.
  2. Line 106-113, It would be more logical to move this paragraph to the front and describe it together with the clinical hazards of influenza viruses.
  3. Line 192, recommended models for phylogenetic tree construction need to be listed.
  4. Line 52-53, “The cleavage site, receptor binding sites (RBS) and glycosylation sites were given special attention”. should be described more clearly.
  5. Figure 5 is not clear and hardly to read.

Author Response

Reviewer 1:

The manuscript [vetsci-3700947], entitled “Emerging highly pathogenic avian influenza H5N1 clade 2.3.4.4b causes neurological disease and mortality in scavenging ducks in Bangladesh” by Prof. Chowdhury, et al., reports the molecular epidemiology and pathology of 22

HPAI H5N1 viruses in unvaccinated scavenging ducks in Bangladesh.

 This is an interesting and important study.

Some concerns are listed below for consideration in revision.

Response: We sincerely thank the Editor for the opportunity to revise our manuscript and for recognizing the significance of our study. We also appreciate the thoughtful comments and suggestions provided. In response, we have carefully revised the manuscript to address each of the concerns raised.

Some specific concerns:

  1. Abstract, the background section in the abstract is excessively lengthy, and the results section fails to specifically present the key data of this study.

Response 1: Revised and edited (Abstract)

  1. Line 106-113, It would be more logical to move this paragraph to the front and describe it together with the clinical hazards of influenza viruses.

Response 2: Revised the whole introduction and lines 113-115.

  1. Line 192, recommended models for phylogenetic tree construction that need to be listed.

Response 3: Included in the text, now in line 232.

  1. Line 52-53, “The cleavage site, receptor binding sites (RBS) and glycosylation sites were given special attention”. should be described more clearly.

Response 4: Revised and described clearly (now in lines 247-249)

  1. Figure 5 is not clear and hardly to read.

Response: Improved Figure 5

Reviewer 2 Report

Comments and Suggestions for Authors

Manuscript “Emerging highly pathogenic avian influenza H5N1 clade 2.3.4.4b causes neurological disease and mortality in scavenging ducks in Bangladesh” by Parvin et al presents valuable findings on the molecular epidemiology, pathology, and evolution of HPAI H5N1 viruses in scavenging ducks in Bangladesh, with a focus on clade 2.3.4.4b. The study is well-structured, methodologically sound, and addresses an important gap in avian influenza surveillance. However, some major points should be addressed to make manuscript refined and fit for the publication.

  1. In introduction, add more comprehensive view on why clade 2.3.4.4b is particularly concerning in the context of global spread and mammalian adaptation e.g. ref 49-50 can be elaborate in the introduction.
  2. Authors should clarify the hypothesis for focusing on scavenging ducks over other poultry and 2.3.4.4b pathogenicity in ducks comparing with earlier clades. It seems to be overgeneralized about pathogenicity of clade 2.3.4.4b is higher than other clades without direct comparison.
  3. Why did authors did not perform histopathology from uninfected ducks as a negative control or if does, add the results
  4. Generate and add a flow chart as a figure how flocks were selected.
  5. Phylogenetic trees for internal genes given in Supplemental Figure S1 should be discussed in the context of reassortment with LPAI viruses.
  6. Authors should compare mortality rates or lesion severity between clade 2.3.2.1a and 2.3.4.4b, if sample size is large enough.
  7. In Table 1: explain why some flocks have high mortality rate irrespective of H5N1 negative?

Minor comments:

  1. In abstract, add novelty of the study like first report of the neurological disease due to H5N1 clade 2.3.4.4b in ducks.
  2. Line 171-172: Clarify the sentence, if sequencing was performed on all 14 positive samples or a subset.
  3. In Fig. 1: No scale bars or annotations for key lesions are mentioned.
  4. In Fig. 2-4: Include magnification scale

Author Response

Reviewer 2:

Manuscript “Emerging highly pathogenic avian influenza H5N1 clade 2.3.4.4b causes neurological disease and mortality in scavenging ducks in Bangladesh” by Parvin et al presents valuable findings on the molecular epidemiology, pathology, and evolution of HPAI H5N1 viruses in scavenging ducks in Bangladesh, with a focus on clade 2.3.4.4b. The study is well-structured, methodologically sound, and addresses an important gap in avian influenza surveillance. However, some major points should be addressed to make manuscript refined and fit for the publication.

Response: We sincerely thank the reviewer for the thoughtful and constructive comments, which have helped us improve the clarity and scientific quality of our manuscript. Below, we provide a detailed response to each of the major points raised. Revisions have been incorporated accordingly in the revised manuscript, and changes are marked for easy identification.

  1. In introduction, add more comprehensive view on why clade 2.3.4.4b is particularly concerning in the context of global spread and mammalian adaptation e.g. ref 49-50 can be elaborate in the introduction.

Response: We appreciate the reviewer’s important observation. In response, we have revised the Introduction to provide a broader context on the global relevance of clade 2.3.4.4b. Specifically, we elaborated on its rapid transcontinental spread, wide avian and mammalian host range, and recent evidence of adaptation to mammalian species, as supported by references 12, 13 and 14. The updated text appears in the revised second and third paragraphs of the Introduction section.

  1. Authors should clarify the hypothesis for focusing on scavenging ducks over other poultry and 2.3.4.4b pathogenicity in ducks comparing with earlier clades. It seems to be overgeneralized about pathogenicity of clade 2.3.4.4b is higher than other clades without direct comparison.

Response: We thank the reviewer for this important comment. In response, we have clarified our study hypothesis in the revised Introduction and Discussion sections (in track change). Scavenging ducks were selected as the focal host due to their unique ecological role in Bangladesh’s poultry production system, where they frequently interact with wild birds, poultry, and shared water bodies—making them important sentinels and amplifiers of AIV transmission (in lines 129-132). Their free-ranging behavior and lack of vaccination increase their susceptibility to infection and potential to serve as reservoirs or mixing vessels for novel AIV strains (in line 450-452).

  1. Why did authors did not perform histopathology from uninfected ducks as a negative control or if does, add the results

Response: Thank you for raising this important point. As part of our field investigation, we performed necropsy on apparently healthy ducks from non-infected flocks. These ducks showed no clinical signs or gross pathological lesions in the respiratory tract or other systemic organs during postmortem examinations. Since no gross abnormalities were observed, we did not proceed with histopathological analysis of these tissues. We have clarified this aspect and briefly noted it in the Results to ensure transparency regarding our control procedures (in lines 330-334).

  1. Generate and add a flow chart as a figure how flocks were selected.

Response: We appreciate the reviewer’s suggestion. As described in the Materials and Methods section (lines 154–156), “a total of 40 scavenging duck flocks were selected based on the presence of at least one reported respiratory or neurological clinical sign. These flocks, ranging in size from approximately 70 to 2500 ducks, were included as part of active and passive surveillance in high-risk areas”. Given the simplicity and straightforward nature of this selection process, we believe that a flow chart would add minimal value and may unnecessarily duplicate existing information. Therefore, we respectfully prefer to keep the current text-based description. However, if the editor recommends its inclusion for clarity, we would be happy to provide a schematic.

  1. Phylogenetic trees for internal genes given in Supplemental Figure S1 should be discussed in the context of reassortment with LPAI viruses.

Response: Thanks for the comment. No such reassortment was observed as now included in lines 363-364.

  1. Authors should compare mortality rates or lesion severity between clade 2.3.2.1a and 2.3.4.4b, if sample size is large enough.

Response: Thank you for this thoughtful suggestion. In our study, only two flocks (SD 4 and SD 13) were confirmed to be infected with clade 2.3.4.4b, exhibiting mortality rates of 17.33% and 0.33%, respectively. The remaining positive flocks were infected with clade 2.3.2.1a viruses, which also showed a broad range of mortality (as detailed in Table 1). Due to the limited number of clade 2.3.4.4b-positive cases, we believe that a statistical comparison of mortality or lesion severity between clades is not robust or conclusive at this stage. Furthermore, variability in flock management, biosecurity practices, and environmental conditions likely contributed to the observed mortality differences. This limitation is now acknowledged and briefly discussed in lines 452-469 of the revised Discussion section.

  1. In Table 1: explain why some flocks have a high mortality rate irrespective of H5N1 negative?

Response: We thank the reviewer for this observation. While a few H5N1-negative flocks showed relatively high mortality, we suspect this may be due to other concurrent infections common in scavenging duck populations, such as duck plague virus, Mycoplasma spp., Pasteurella multocida, or Riemerella anatipestifer. However, as the primary objective of this study was to investigate the presence and impact of HPAI H5N1, further diagnostic workup for other pathogens was not conducted.

Minor comments:

  1. In abstract, add novelty of the study like first report of the neurological disease due to H5N1 clade 2.3.4.4b in ducks.

Response: Included in line 59

  1. Line 171-172: Clarify the sentence, if sequencing was performed on all 14 positive samples or a subset.

Response: Thanks for this comment. We were able to sequence 7 samples, and selection was made based on the CT values now added on line 203

  1. In Fig. 1: No scale bars or annotations for key lesions are mentioned.

Response: Added.

  1. In Fig. 2-4: Include magnification scale

Response: Added

Reviewer 3 Report

Comments and Suggestions for Authors

Manuscript ID: vetsci-3700947 was submitted by Rokshana Parvin et al. for Veterinary Sciences.

This study provides insights into HPAI H5N1 clade 2.3.4.4b neuropathogenicity in the domestic ducks, highlighting an emerging public health threat in Bangladesh. However, conclusions about HPAI H5N1 clade 2.3.4.4b’s increased virulence require more robust molecular and pathological evidence. Addressing these concerns will significantly strengthen the manuscript’s impact.  

  1. High mortality (e.g., 50% in SD32; Table 1) occurred in H5N1-negative flocks, why? Co-infections (e.g., duck plague virus, bacterial pathogens) were not ruled out, potentially confounding pathology and mortality data.
  2. Table 2 compares isolates only to A/Goose/Guangdong/1/1996. Contemporary 2.3.4.4b strains (e.g., 2022–2024 Asian lineages) should be included to contextualize HA/NA substitutions.
  3. The asserted "increase in pathogenicity" of clade 2.3.4.4b (vs. historical asymptomatic cases) lacks parallel data from contemporaneous asymptomatic ducks. Comparison to local surveillance data or experimental infection studies is needed.
  4. In figures 2–4,the Histopathological lesions of control group duck should be present.
  5. Mutation analysis of the HA protein in clade 2.3.4.4b viruseswas carried out, but whether T156A,S141P,E193R/K relulate the antigenic drift and receptor-binding adaptation still was unclear.
  6. Suspected duplication in flock IDs (e.g., SD33 vs. SDS1/SDS2; Table 1) requires verification.

Author Response

Reviewer 3:

Manuscript ID: vetsci-3700947 was submitted by Rokshana Parvin et al. for Veterinary Sciences.

This study provides insights into HPAI H5N1 clade 2.3.4.4b neuropathogenicity in the domestic ducks, highlighting an emerging public health threat in Bangladesh. However, conclusions about HPAI H5N1 clade 2.3.4.4b’s increased virulence require more robust molecular and pathological evidence. Addressing these concerns will significantly strengthen the manuscript’s impact.  

  1. High mortality (e.g., 50% in SD32; Table 1) occurred in H5N1-negative flocks, why? Co-infections (e.g., duck plague virus, bacterial pathogens) were not ruled out, potentially confounding pathology and mortality data.

Response: we have now discussed the mortality issues in lines 452-469.

  1. Table 2 compares isolates only to A/Goose/Guangdong/1/1996. Contemporary 2.3.4.4b strains (e.g., 2022–2024 Asian lineages) should be included to contextualize HA/NA substitutions.

Response: Thank you for this insightful comment. We acknowledge the value of including contemporary 2.3.4.4b reference strains for better contextualization of amino acid substitutions. In our study, such comparisons were already provided in the Supplementary Figure S1, where representative 2.3.4.4b strains from 2022–2024 circulating in Asia (including isolates from Korea and Vietnam) were used in the phylogenetic and mutation analyses. Since the primary intent of Table 2 was to summarize key molecular features relative to the ancestral A/Goose/Guangdong/1/1996 H5 lineage as a historical baseline, we did not duplicate the comparative data from the more recent strains.

  1. The asserted "increase in pathogenicity" of clade 2.3.4.4b (vs. historical asymptomatic cases) lacks parallel data from contemporaneous asymptomatic ducks. Comparison to local surveillance data or experimental infection studies is needed.

Response: We appreciate the reviewer’s critical point. We agree that the assertion regarding increased pathogenicity of clade 2.3.4.4b is based on field observations and histopathological findings in two flocks, and not on a controlled comparative study. Our conclusion was drawn from the notable neurological and systemic lesions observed in naturally infected ducks of clades 2.3.4.4b. Accordingly, we have revised the relevant statements in the Discussion to more cautiously interpret the findings. We now emphasize that our findings suggest a potentially increased pathogenicity of clade 2.3.4.4b in ducks and recommend that further controlled studies and broader surveillance, including asymptomatic populations are needed to validate this observation (In lines 478-481).

  1. In figures 2–4, the Histopathological lesions of control group duck should be present.

Response: Thank you for this suggestion. As mentioned in lines 330–334 of the revised manuscript, we performed necropsies on apparently healthy ducks from H5N1-negative flocks. These ducks exhibited no gross pathological changes, and therefore histopathological examination was not conducted. Since no histological slides were generated from these negative controls, we are unable to include comparative images in Figures 2–4. We have clarified this rationale in the Materials and Methods sections to provide transparency regarding our approach.

  1. Mutation analysis of the HA protein in clade 2.3.4.4b viruses was carried out, but whether T156A, S141P, E193R/K relulate the antigenic drift and receptor-binding adaptation still was unclear.

Response: Thank you for this insightful comment. As noted in Table 2, we observed mutations such as T156A, S141P, and E193R/K in some of the strains. These substitutions are located within or near known antigenic sites and the receptor-binding domain and have been reported in previous studies to be associated with antigenic variation or altered receptor affinity (added in lines 385-389). However, we fully agree with the reviewer that their precise role in antigenic drift and receptor-binding adaptation in our isolates remains speculative without supporting functional assays.

We have now clarified this point in the Discussion section by stating that the potential phenotypic impacts of these mutations require further investigation through in vitro receptor-binding assays, cell culture studies, or animal models to confirm their functional significance (added in lines 393-396).

  1. Suspected duplication in flock IDs (e.g., SD33 vs. SDS1/SDS2; Table 1) requires verification.

Response: Sorry for this unexpected mistake. Table 1 has now been revised, and duplication was removed (Table 1).

Reviewer 4 Report

Comments and Suggestions for Authors

Very well designed study, using the golden standard methods of the field. Very ell written manuscript; not just the scientific content is correcect, but both the text and the figures are well edited and designed.

My only minor concern is the labelling of NA phylog. tree as 2.3.4.4b and 2.3.2.1a. I understand, what you mean, but it is misleading in this form. Please make it clear in the capture, that this clades are referring the HA (!) segments, and these clades with angle brackets representing monophyletic groups of viruses containing similar HA and NA segments as well.

Please contact to the Editor about the correct and proportinal sizes of the figures and tables. I would reduce the size of the histopathological photos and definitelly the tables. Nevertheless I consider the phylogenetic trees of the internal segments so important, to incorporate them from the suppl. data to the article.

Author Response

Reviewer 4:

Very well designed study, using the golden standard methods of the field. Very ell written manuscript; not just the scientific content is correcect, but both the text and the figures are well edited and designed.

My only minor concern is the labelling of NA phylog. tree as 2.3.4.4b and 2.3.2.1a. I understand, what you mean, but it is misleading in this form. Please make it clear in the capture, that this clades are referring the HA (!) segments, and these clades with angle brackets representing monophyletic groups of viruses containing similar HA and NA segments as well.

Response: Thank you for pointing this out. We agree that the current labeling could lead to confusion, as 2.3.4.4b and 2.3.2.1a refer to HA clades, not NA gene classifications. In response, we have revised the figure legend of the NA phylogenetic tree to clearly state that the clade names shown (e.g., 2.3.4.4b and 2.3.2.1a) refer to the hemagglutinin (HA) clade designation and are used here to indicate the overall genetic background of the viruses, which share both HA and NA segment relationships (in lines 352-254).

Please contact to the Editor about the correct and proportinal sizes of the figures and tables. I would reduce the size of the histopathological photos and definitelly the tables. Nevertheless I consider the phylogenetic trees of the internal segments so important, to incorporate them from the suppl. data to the article.

Response: Thanks for your suggestions. Actually, we have submitted all figures separately in the system. Hope they will upload it during the final published copy.

Round 2

Reviewer 2 Report

Comments and Suggestions for Authors

Authors have addressed all the potential queries.